# An Ultrasonic Motor Using a Carbon-Fiber-Reinforced/Poly-Phenylene-Sulfide-Based Vibrator with Bending/Longitudinal Modes

**DOI:** 10.3390/mi13040517

**Published:** 2022-03-26

**Authors:** Zhaochun Ding, Wentao Wei, Keying Wang, Yumeng Liu

**Affiliations:** 1School of Mechanical Engineering, Shandong University, Jinan 250061, China; weiwentao@mail.sdu.edu.cn (W.W.); wangkeying@mail.sdu.edu.cn (K.W.); 2School of Control Science and Technology, Shandong University, Jinan 250061, China; liuyumeng@mail.sdu.edu.cn

**Keywords:** carbon-fiber-reinforced poly phenylene sulfide (PPS/CF), anisotropic material, Young’s moduli, polymer, ultrasonic motor

## Abstract

In this study, a linear ultrasonic motor (USM) with carbon-fiber-reinforced/poly-phenylene-sulfide (PPS/CF) was developed and the feasibility of using PPS/CF to achieve a lightweight USM was tested. Here, anisotropic Young’s moduli of PPS/CF possibly enhance the driving force when the slider’s moving direction is orthogonal to the carbon-fibers’ filling direction. Further, PPS/CF’s low density may help avoid excessive enhancement in weight. Initially, we measured anisotropic Young’s moduli of PPS/CF, and determined the vibration modes, configuration, and dimensions of the PPS/CF vibrating body through finite element analysis. Subsequently, we fabricated a 45.7-mm-long 30-mm-diameter vibrator to form a linear motor. Finally, we evaluated the load characteristics of the PPS/CF-based motor and made comparisons with isotropic-material-based USMs. At 30.2 kHz frequency, the PPS/CF-based vibrator worked in the 2nd bending and 2nd longitudinal modes as predicted. The PPS/CF-based motor yielded the maximal thrust, no-load speed, and maximal output power of 392 mN, 1103 mm/s, and 62 mW, respectively. Moreover, the thrust force density and power density reached 20.3 N/kg and 3.2 W/kg, respectively, which were relatively high compared to those of the PPS- and aluminum-based USMs with the same vibration modes and similar structures. This preliminary study implies PPS/CF’s feasibility for achieving lightweight USMs, and provides a candidate material for designing micro/meso USMs.

## 1. Introduction

Ultrasonic motors (USMs) convert electrical energy into mechanical energy on the basis of the inverse piezoelectric effect and achieve actuation via friction [1,2,3,4]. Compared to electromagnetic motors, they offer the advantages of a quick response, self-locking at the power-off stages, and the absence of electromagnetic radiation [5,6,7], which make USMs applicable to some special areas (e.g., auto-focus lenses in optical instruments or digital cameras [8]). To accelerate their practical usage in wider fields, a lightweight design is desirable for USMs, as this contributes to reducing the entire instrument’s weight and improving the controllability [9,10]. USMs are basically composed of vibrators and sliders [11], with the vibrators having a fundamental effect on vibration properties [1,8,12]. In general, the vibrators incorporate piezoelectric materials and vibrating bodies [1,8,13], with the piezoelectric materials (lead zirconate titanate (PZT) in most cases) acting as excitors [1,14], and meanwhile, the vibrating bodies achieve modal degeneration (i.e., gathering the resonance frequencies of different vibration modes) and enlarge vibration amplitude [15]. Since it is not cost-efficient to use PZT plates with non-uniform shapes [16], many technicians focus on modifying vibrating bodies to decrease the weight of USMs [17,18]. For instance, structural optimization has been heavily adopted to remove redundant portions [19]. However, modal degeneration makes it difficult to change the vibrators’ configurations to a great extent [20], and consequently limits the motors’ weights [2,20]. On the other hand, some literature studies have discussed how the vibrating bodies’ material constants affect USMs’ performance, and have contributed some basic understanding about several typical materials’ suitability for USMs [12,21,22]; these allow us to infer that selection of the vibrating body’s material is an alternative approach to achieve lightweight USMs.

Metals, ceramics, and polymers are commonly-used engineering materials [8,23]. To date, most USMs have adopted metal as their vibrating bodies [8], but large densities of metals (except aluminum) become an obstacle to achieving a lightweight design [23]. Fine ceramics (e.g., alumina, silicon carbide, and aluminum nitride) enable the motors to produce large mechanical output at low voltage, but ceramics’ moderate densities (~4 × 10^3^ kg/m^3^) limit further reduction of USMs’ weights [24]. In addition, ceramics’ fragility becomes an obstacle to making fine ceramics’ vibrating bodies with complex structures [8]. Polymers generally provide unique properties of low densities (<2 × 10^3^ kg/m^3^) [23,25], but most of them are incapable of propagating intensive vibration for large mechanical loss [12]. As an exception, poly phenylene sulfide (PPS) interestingly has low mechanical loss (e.g., its damping coefficient is about two orders of magnitude lower than those of commonly-used polymers) [26], indicating the potential of using PPS as the vibrating bodies of USMs [1]. Wu et al. utilized PPS to form several ring-shaped motors with different modes [12,21,27], and Cao et al. analyzed PPS’ vibration properties with nonlinear dynamic models [25]. Although PPS-based motors have small weights, their mechanical outputs require further enhancement [25,27]. 

In this study, we employed a new type of engineering polymer, i.e., carbon-fiber-reinforced poly phenylene sulfide (PPS/CF), as the vibrating body, and tested the feasibility of achieving lightweight USMs’ without excessively deteriorating their mechanical outputs. Herein, we assume PPS/CF to be a strong candidate material for the following reasons. 

(1) Low density: PPS/CF has low density (1.69 × 10^3^ kg/m^3^), although numerous carbon fibers are filled in the base material (PPS, whose density is 1.36 × 10^3^ kg/m^3^); this contributes to achieving lightweight USMs [28,29]. 

(2) Anisotropic Young’s moduli: The anisotropy originates from the fact that filling the carbon fibers makes the Young’s modulus in the filling direction higher than the values in the other directions [30,31]. If the filling direction is set perpendicularly to the sliding direction of the motor, the relatively high Young’s modulus in the filling direction possibly causes the PPS/CF-based vibrator to produce a large driving force. In the meantime, the relatively low Young’s modulus orthogonal to the filling direction likely causes small stiffness in the sliding direction, which can help avoid significant reductions in the vibration velocity. Despite this, anisotropy is undesirable for most mechatronic systems, for USMs, it brings the possibility of improving their performance on the condition that the carbon fibers’ filling direction are optimally set. 

(3) Low mechanical loss. PPS/CF inherits the low mechanical loss of PPS [29], indicating the feasibility of making intensive ultrasound propagatable in the PPS/CF-based vibrator. This is a feature shared by PPS/CF and other types of reinforced polymers. 

The above discussion indicates that PPS/CF exhibits the potential to be used as the vibrating bodies of USMs. Further, to the best of our knowledge, PPS/CF has not been used as the vibrating bodies of USMs, except for in [29]. Therefore, it would be meaningful to exploit PPS/CF-based motors to examine whether the usage of this new material facilitates the realization of lightweight USMs. 

In this study, we developed a PPS/CF bar-shaped vibrator with the longitudinal/bending modes to form a linear motor and conducted a preliminary investigation on its structural design, vibration properties, and performance assessment. The rest of this paper is organized as follows. After Section 1 introduces the background and motivation, Section 2 demonstrates the measurement of PPS/CF’s anisotropic Young’s moduli. Then, Section 3 illustrates the PPS/CF-based vibrator’s configuration and its working principle. Next, Section 4 and Section 5 show the vibration properties and the load characteristics, respectively, both of which are compared with the performances of several isotropic material-based USMs. Section 6 concludes this paper.

## 2. Measurement of PPS/CF’s Anisotropic Young’s Moduli

To conduct modal degeneration via finite element analysis (FEA), it was necessary to measure PPS/CF’s anisotropic Young’s moduli. Since PPS/CF fills the carbon fibers in the base material, its anisotropic Young’s moduli are commonly expressed with the orthogonal model [32]:
(1)$$E=\left[\begin{array}{cccccc} E_{xx} & E_{xy} & E_{xz} & 0 & 0 & 0\\ & E_{xx} & E_{xz} & 0 & 0 & 0\\ & & E_{zz} & 0 & 0 & 0\\ & & & G_{xx} & 0 & 0\\ & sym. & & & G_{xx} & 0\\ & & & & & G_{zz} \end{array}\right]\;=\left[\begin{array}{cccccc} E_{xx} & E_{xx}-2G_{zz} & E_{zz}-2G_{xx} & 0 & 0 & 0\\ & E_{xx} & E_{zz}-2G_{xx} & 0 & 0 & 0\\ & & E_{zz} & 0 & 0 & 0\\ & & & G_{xx} & 0 & 0\\ & sym. & & & G_{xx} & 0\\ & & & & & G_{zz} \end{array}\right].$$

Here, *E_xx_* and *G_xx_* are tensile and shear moduli orthogonal to the carbon-fibre’s filling direction, respectively, and *E_zz_* and *G_zz_* are tensile and shear moduli along the carbon-fibres’ filling direction, respectively and these values are estimated as
(2)$$\{\begin{array}{c} E_{xx}=\rho c_{L-x}^{2}\\ G_{xx}=\rho c_{T-x}^{2}\\ E_{zz}=\rho c_{L-z}^{2}\\ G_{zz}=\rho c_{T-z}^{2} \end{array}.$$

Here, *ρ* represents PPS/CF’s density (1.69 × 10^3^ kg/m^3^). *c_L−x_* (*c_L−z_*) and *c_T−x_* (*c_T-z_*) denote the transverse and longitudinal wave speeds, respectively, when the propagating directions are orthogonal (parallel) to the filling direction [32].

Figure 1 illustrates the schematic for measuring *c_L−z_*. In this study, we used a PPS/CF sample with properties similar to Toray’s products [33], and this sample was created by conventional machining. A 30.3-mm-long 30-mm-diameter PPS/CF bar was clamped with a pair of longitudinal ultrasonic probes (A194S-RB, Olympus, Tokyo, Japan), with a probe sending out a continuous signal while the other one received the signal. We recorded the sending-out and initial-arriving instances, whose interval was regarded as the time period of propagation Δ*t*, supposing that the sample’s length equalled Δ*l_sample_*, *c_L−z_* = Δ*l_sample_*/Δ*t*. Similarly, *c_T−z_* was measured with a pair of transverse wave-type probes (A191S-RB, Olympus, Tokyo, Japan). Figure 2a shows the waveform in the time domain. Observably, the first arrivals occurred at 6.44 and 13.45 μs for the longitudinal and transverse waves, respectively. Following these procedures, we prepared some other samples with different lengths, recorded the lengths and time periods, and calculated the wave speeds by fitting curves. As shown in Figure 2b, the longitudinal and transverse wave speeds were 3062 and 1670 m/s, respectively, in the filling direction. In addition, we prepared another sample to measure *c_L−x_* and *c_T−x_* by slicing the PPS/CF bar parallelly to the filling direction. The longitudinal and transverse wave speeds were 2639 and 1429 m/s, respectively, when the waves propagated orthogonally to the filling direction. The Young’s moduli of PPS/CF were calculated according to Equation (1) and the results are listed in Table 1.

## 3. Configuration and Working Principle

### 3.1. Configuration

As Figure 3a illustrates, the PPS/CF-based vibrator consisted of a vibrating body, four PZT plates, and a driving foot. The PPS/CF vibrating body contained rectangular and cylindrical parts, whose dimensions are shown in Figure 3b. The rectangular part was 30 mm in diameter and 10 mm in thickness and it was tightly pressed with a 60-mm-long 60-mm-wide 10-mm-thick cover plate. The rectangular part is 13 mm in length (and width) and its length was varied to achieve modal degeneration. Four pieces of PZT plates that were 18 mm in length, 10 mm in width, and 0.2 mm in thickness (P4, Hongsheng Acoust.) were bonded onto the rectangular part. The alumina driving foot, which was 2 mm in diameter, was inserted into the vibrating body, and it drove the slider frictionally. 

The PPS/CF-based vibrator’s resonance frequencies were calculated via FEA (software: ANSYS 14.5, ANSYS Corporation, Canonsburg, PA, USA). Both the PZT plates and vibrating bodies were meshed with the element type of SOLID5, which is capable of processing anisotropic materials with orthogonal models. Figure 3c illustrates how the resonance frequencies of various modes depended on the rectangular part’s length *l*. It can be observed that, at *l* = 40 mm, the 2nd longitudinal (L2) and 2nd bending (B2) modes had identical resonance frequencies (~30 kHz). Although the 3rd longitudinal and 2nd bending modes’ resonance frequencies were gathered at *l* = 21 mm, the PPS/CF vibrating body provided insufficient length for bonding the PZT plates. On the other hand, modal degeneration was achievable at larger *l*, which is, however, a main obstacle to decreasing the weight. Thus, we set *l* to 35.7 mm and prototyped the PPS/CF-based vibrator. For comparison, we made PPS- and aluminium-based vibrators, which worked in the same modes and had similar structures to the PPS/CF-based vibrator. It should be noted that, for the PPS- and aluminium-based vibrators, *l* equalled respectively 36 and 58 mm, which were estimated by modal degeneration through FEA. Figure 4. shows images of these prototypes. Further, a 400-mm-long linear guide was prepared as the slider. An aluminium sheet was glued onto the slider’s bottom surface as the friction material.

### 3.2. Working Principle

Figure 3d illustrates the method of applying voltage. The B2 vibration was excited with two channels of out-of-phase voltages (*V*_0_cos(*ω**t*) and -*V*_0_cos(*ω**t*), where ω and *t* are respectively the angular frequency and time), while the L2 vibration was generated with two channels of in-phase voltages (*V*_0_cos(*ω**t* + *φ*) and *V*_0_cos(*ω**t* + *φ*), where *φ* means the phase). The operating sequence at *φ* = 90° were as follows (see Figure 3e).

(i) The z-axis vibration displacement and the *x*-axis vibration velocity reached the peak values along the +*z* and +*x* axes, respectively. 

(ii) The longitudinal vibration returned to its original position. The bending vibration had the maximal displacement, but the *x*-axis vibration velocity was zero. 

(iii) The *z*-axis vibration displacement and the x-axis vibration velocity were maximal in the -*z* and -*x* axes, respectively. 

(iv) The vibration state was inverse to that in step (ii). 

In step (i), the vibrator accelerated the slider as its *x*-axis vibration velocity exceeded the sliding speed. In contrast, the slider was decelerated in the other steps. At *φ* = −90°, the operating sequence became (i)→(iv)→(iii)→(ii), and consequently, the sliding direction was inverse to that at *φ* = 90°.

## 4. Vibration Properties

### 4.1. Admittance Characteristics

First, the admittance characteristics of the PPS/CF-, PPS-, and aluminum-based vibrators were explored based on the equivalent circuit model [6,8,24,33]. Initially, the admittance curves were measured with an impedance analyzer (4294A, Agilent, Santa Clara, CA, USA) to obtain the resonance frequency (*f_r_*), anti-resonance frequency (*f_a_*), mechanical quality factor (*Q*), motional admittance (*Y_m_*_0_), and clamped capacitance (*C_d_*). Subsequently, other parameters, i.e., the electromechanical coupling factor (*k*), motional resistance (*R_m_*), motional inductance (*L_m_*), and motional capacitance (*C_m_*), were estimated with the following equations [24,34]:(3)$$k=\sqrt{\frac{f_{a}^{2}-f_{r}^{2}}{f_{a}^{2}}},$$
(4)$$R_{m}=\frac{1}{Y_{m0}},$$
(5)$$C_{m}=\frac{Y_{m0}}{2\pi f_{r}\times Q},$$
and
(6)$$L_{m}=\frac{Q}{2\pi f_{r}\times Y_{m0}}.$$

Note that *Q* factors were measured at low vibration-amplitude region (the voltage was 1 V, the maximal value providable with the impedance analyser). Table 2 shows the results. First, the working frequencies were notably lower for the PPS/CF- and PPS-based vibrators than for the aluminium-based one, because PPS/CF and PPS have smaller Young’s moduli than aluminium [29]. Second, the *Q* factor of the PPS/CF-based vibrator was comparable to that of the PPS/CF-based one, and meanwhile, the relatively high Young’s modulus of PPS/CF brought a relatively large electromechanical coupling factor to the PPS/CF-based vibrator compared to the PPS- based one. Third, the aluminium-based vibrator’s *k* was about twice that of the PPS/CF-based vibrator, despite aluminium’s much larger Young’s modulus.

### 4.2. Vibration Velocity Distribution

Then, the vibration velocity distributions were investigated via interferometric measurement. The vibrators were fixed onto a base board with a cover plate. The voltage applied to each vibrator was set to 10 V (note that, in this paper, the voltage as well as the vibration velocity are indicated as zero-to-peak values), and the working frequencies were 30.3, 23.1, and 59.4 kHz for the PPS/CF-, PPS-, and aluminium-based vibrators, respectively. The longitudinal vibration velocity was measured with an in-plane vibrometer (LV-IS01, Sunny Optical Corp, Zhejiang, China) while the bending vibration velocity was measured with an out-of-plane vibrometer (LV-S01, Sunny Optical Corp, Zhejiang, China). The vibrometers were arranged on a stage capable of moving along the *z* axis to accomplish the automatic scanning. 

Figure 5a illustrates the vibration velocity distributions of the PPS/CF-based vibrator. As predicted, the L2 and B2 vibrations existed on the vibrator. The longitudinal and bending vibration velocities reached the maximal values of 63 and 88 mm/s, respectively. Figure 5b shows the vibration velocity distributions of the PPS-based vibrator, which did not greatly differ from those of the PPS/CF-based vibrator. However, the maximal vibration velocity achievable with the PPS-based vibrator was 1.3 times that of the PPS/CF-based one probably because of PPS’s smaller Young’s modulus. As shown in Figure 5c, the aluminium-based vibrator took longer to gather the resonance frequencies of L2 and B2 modes owing to aluminium’s high Young’s modulus [29], and its vibration velocity at the free end was smaller than those of the PPS/CF- and PPS-based vibrators. It should be noted that the position of bonding PZT plates should be near the nodes of both longitudinal and bending modes to efficiently excite the vibrations. 

### 4.3. Vibration Velocity versus Driving Voltage

Finally, the vibration velocity of the bending mode was experimentally explored as a function of the driving voltage. The working frequencies of the PPS/CF-, PPS-, and aluminium-based vibrators were set to, respectively, 30.3, 23.1, and 59.4 kHz. Clearly, the vibration velocities monotonically became higher with increasing voltage. As Figure 6a,b show, the maximal vibration velocities of the PPS/CF- and PPS-based vibrators were, respectively, 1152 and 1396 mm/s at 180 and 190 V (referred to as the critical voltage), where the fracture was observable on the PZT plates. Figure 6c shows that, for the aluminium-based vibrator, the vibration velocity reached 276 mm/s at the critical voltage of 160 V. During performance evaluation of the PPS/CF-, PPS-, and aluminium-based motors, we set the driving voltage to, respectively, 160, 160, and 140 V, which deviated from their critical voltages to avoid fracture.

## 5. Motor Performance

### 5.1. Load Characteristics

Figure 7a schematically illustrates the testbed for measuring the load characteristics, with Figure 7b showing an image of the testbed. The thrust force was estimated by pulling up weights, while the sliding speed was measured with a laser displacement sensor (WT53R, Witmotion Technology Corporation, Guangzhou, China). Here, two guides were used to enable the slider to move parallelly to the base. As shown in Figure 7c, both the vibrator and the preload-applying mechanism were arranged vertically to the base to suppress the impact of gravity-induced effects on the load characteristics. A coil spring was compressed to apply the preload, which was able to be adjusted by rotating the screw.

Initially, we explored how the sliding speed depended on the phase between the voltages for exciting longitudinal and bending vibrations. The results of the PPS/CF-based motor at 60 V voltage and ~30.3 kHz frequency are plotted in Figure 8a. At 70° and −110°, the sliding speed reached the maximal values in opposite directions. Moreover, when the phase monotonically decreased from 70° to −110°, the sliding speed provided a gradual reduction from 328 mm/s to zero, and then, a gradual enhancement to 323 mm/s in the inverse direction; the dead zone (which indicates the phase range corresponding to zero speed) was ~10°. These results imply the PPS/CF motor’s capability to adjust the sliding speed. Figure 8b,c show how the vibration velocities depended on the phase for the PPS- and aluminum-based motors, respectively, where the working frequencies were ~23.1 and ~59.4 kHz. It is observable that the dependencies of these motors had identical tendencies. Further, the phase ranges corresponding to the dead zones were almost the same for the PPS/CF- and PPS-based motors, and both of them are smaller than that for the aluminum-based one. The phases corresponding to the maximal vibration velocities slightly deviated from the theoretical values (±90°), probably because of imperfect fabrication and/or assembly [3], which are generally unavoidable for USMs. 

Then, we investigated the load characteristics of the PPS/CF-based motor, where the voltage and working frequency were set to respectively 160 V and ~30.2 kHz. Figure 9a–c shows the preloads applicable to the motor at different voltages that were explored for the PPS/CF-, PPS-, and aluminum-based motors, respectively. When the motors worked under the green area (shown as (I) in Figure 9), where the voltage was higher and the preload was not sufficient, they provided continuous movement. When the preloads became higher (in the yellow area (II)), the motors generally provided continuous movement, but some sticks were observed. While the preloads further increased (in the red area (III)), the motors were not able to have continuous movement. The applicable preload being higher for the aluminum-based motor may have originated from its larger motional admittance [11]. Further, the PPS/CF-, PPS-, and aluminum-based motors could not work at >180, >180, and >140 V, respectively, because their PZT plates easily fracture at these ranges. According to the results in Figure 9, we set the preloads to 1, 3, and 5 N for the PPS/CF motor, 1, 3 and 3.6 N for the PPS one, and 4 and 9.8 N for the aluminum one. 

Figure 10a plots the variation in sliding speed versus the thrust force. At 1 N preload, the no-load sliding speed and maximal thrust force reached 1103 mm/s and 68.6 mN, respectively. When the preload increased to 3 and 5 N, the thrust force increased to 235.2 and 392.0 mN. As shown in Figure 10b,c, the PPS/CF-based motor yielded the maximal output power and maximal efficiency of, respectively, 73.5 mW and 17.1% at a moderate preload of 3 N. Figure 10d–f illustrate, respectively, how the sliding speed, output power, and efficiency of the PPS-based motor depend on the thrust force when the voltage was 160 V and the working frequency was ~23.1 kHz. Since the preload applicable to the PPS-based motor was smaller than that to the PPS/CF-based one, we set the preloads to 1, 3, and 3.6 N. It should be noted that the maximal preload applied to the PPS-based motor could not reach 5 N (the value applicable to the PPS/CF-based motor) as the PPS-based motor is incapable of moving as a consequence of large vibration reduction [29]. Both the maximal thrust force and maximal output power were lower for the PPS- than for the PPS/CF-based motor. Figure 10g–i respectively plot, respectively, the variations in sliding speed, output power, and efficiency of the aluminum-based motor against the thrust force at 140 V voltage and ~59.2 kHz. It can be seen that the aluminum-based motor was able to work at larger preloads, with the thrust force, no-load sliding speed, maximal output power, maximal efficiency being 529 mNm, 243 mm/s, 37 mW, and 13%, respectively. 

Figure 11a–d plots, respectively, the variations in maximal thrust force, no-load sliding speed, maximal output power, and maximal efficiency against the voltage under different preloads. First, when the preload was 1 N, at the voltage of <40 V, there existed small thrust force due to insufficient driving force. Then, the thrust force became higher with increasing voltage but leveled off at certain voltages. The no-load sliding speed gradually became higher with increasing voltage. The maximal output power corresponding to 3 N preload exceeded those corresponding to the other preloads because this moderate preload avoids not only insufficient driving force (which generally exists at small preloads) but also excessive frictional loss (which is commonly caused by an unnecessarily large preload) [21,27]. The efficiency corresponding to the preload of 1 N increased in the voltage range of 40–80 V, and decreased when the voltage was over 120 V. Figure 11e–h plot, respectively, the maximal thrust force, no-load sliding speed, maximal output power, and maximal efficiency of the PPS-based motor as functions of voltage at different preloads. The no-load sliding speed achievable with the PPS-based motor was 1261 mm/s, larger than that achievable with the PPS/CF-based one (1102 mm/s). As shown in Figure 10i–l, the aluminum-based motor had larger thrust force, but smaller sliding speed and smaller output power than the PPS/CF- and/or PPS-based ones. The maximal efficiencies of these motors did not show large differences probably because the contacting surfaces between the vibrators and sliders are identical [8].

Then, we explored the heat generation of the three motors working at their maximal voltages. The room temperature was set to 20 °C and the temperatures of the motors’ surfaces were measured with a thermal imager (UTi12OT, Uni-trend Technology Corporation, Guangzhou, China). Initially, an example of temperature rise of the PPS/CF-based motor was obtained, where the vibrator, as shown in Figure 12a, recorded a higher temperature than the other parts. Then, the temperature corresponding to the vibrator zone was averaged and the results were plotted, as shown in Figure 12b as a function of time. After 40-min of continuous excitation, the temperatures of the PPS/CF- and PPS-based motors increased by ~14 and ~13 °C, respectively, which were not markedly different from that of the aluminum-based one (~11 °C). 

### 5.2. Performance Comparison

Finally, the PPS/CF-based motor’s performance was compared with those of the PPS- and aluminum-based motors. The results listed in Table 3 allow us to infer the following: 

(1) The lengths were smaller for the PPS/CF- and PPS-based motors than for the aluminum-based one when the three motors worked in the same modes and had similar structures. In addition, low densities as well as the small dimensions (volumes) led to relatively small weights for the PPS/CF- and PPS-based motors [35,36]. 

(2) Since the carbon fibers’ filling direction was perpendicular to the sliding direction, as predicted, the PPS/CF-based motor produceds relatively high thrust force compared to the PPS-based one [12]. On the other hand, the higher Young’s moduli of PPS/CF did not cause far smaller sliding speed for the PPS/CF- than for the PPS-based motor [27]. 

(3) Since the usage of carbon fibers can enhance the thrust force and/or output power without excessively increasing the weight, the PPS/CF-based motor produced relatively high thrust force density and high power density compared to the PPS-based one. Meanwhile, owing to lower density and smaller volume, the PPS/CF-based motor’s thrust force density and power density exceeded those of the aluminum-based motor. 

(4) Compared to the PPS/CF-based motor with orthogonal bending (B2/B2) modes, the PPS/CF-based motor reported in this study had relatively large thrust force, sliding speed, and output power probably because the longitudinal vibration has relatively strong electromechanical coupling.

In short, PPS/CF can bring a lightweight quality to USMs; this validates the effectiveness of our proposal.

## 6. Conclusions

This article presents, to our knowledge, the first report on the proposal, design, and performance assessment of the PPS/CF-based motor. Through analytical and experimental investigations, we have drawn the following conclusions: 

(1) A new approach of using anisotropic polymers as the vibrating body was proposed to design lightweight USMs. 

(2) PPS/CF’s relatively high Young’s moduli enable the PPS/CF-based motor to produce a relatively large resonance frequency and a large electromechanical coupling factor compared to the PPS-based vibrator.

(3) The PPS/CF-based motor exhibited relatively high thrust force and high output power compared to the PPS-based one, while the no-load sliding speeds of these two motors were comparable; this implies there was a carbon-fiber-induced positive effect on the polymer-based USMs.

(4) The PPS/CF-based motor yielded a thrust force density and power density of 20.3 N/kg and 3.2 W/kg, respectively, which was relatively high compared to those of the PPS- and aluminum-based ones working in identical modes and having similar structures; these results infer that anisotropic polymer can bring a lightweight design to USMs to some extent. 

These conclusions verify the feasibility of achieving lightweight USMs by using PPS/CF, and indicate PPS/CF’s potential to be used for micro/meso-sized USMs [37]. It is worth mentioning that the PPS/CF-based motor developed in this study does not provide performance comparable to the commercially-available ultrasonic motors, e.g., Shinsei’s product [38], as some factors, such as the bonding materials, the structure, and the friction materials, were not fully considered, but we assume that this problem does not hamper this work’s contribution with regards to applying the polymer to USMs and showing basic designing procedures. In the future, we will discuss how carbon-fibers’ filling direction affects the performance of other types of USMs and accordingly conduct structural optimization. It is also worthwhile to investigate the characteristics of the stepping operation of the PPS/CF-based motor to broaden its application areas.

## Figures and Tables

**Figure 1 micromachines-13-00517-f001:**
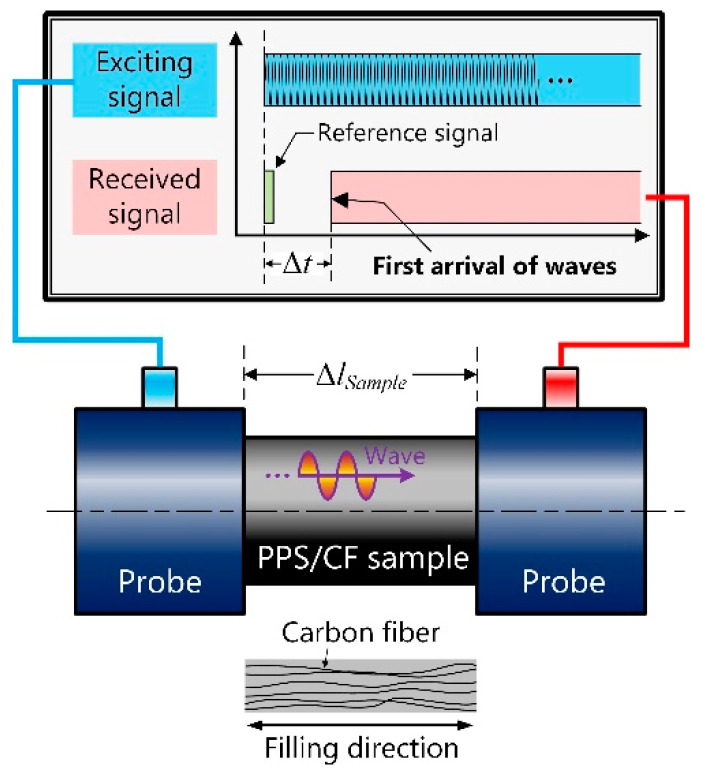
Schematic of measuring PPS/CF’s anisotropic Young’s moduli.

**Figure 2 micromachines-13-00517-f002:**
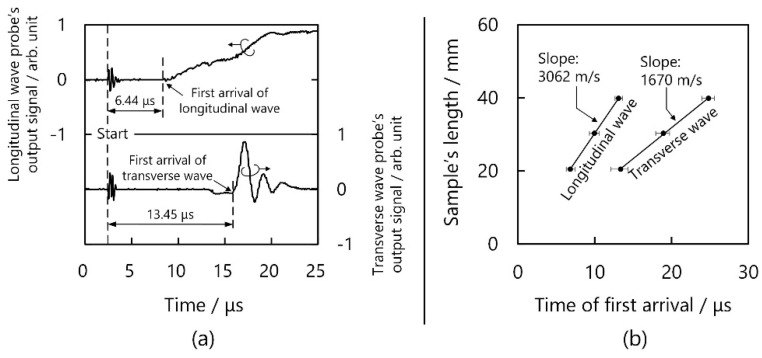
Experimental results. (**a**) Waveform in the time domain and (**b**) calculation of longitudinal and transverse wave speeds.

**Figure 3 micromachines-13-00517-f003:**
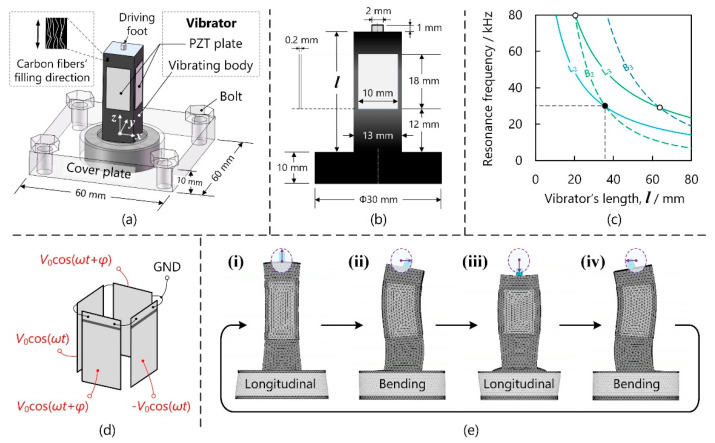
PPS/CF-based vibrator. (**a**) Configuration, (**b**) dimensions, (**c**) modal degeneration, (**d**) method for applying voltage, and (**e**) working principle.

**Figure 4 micromachines-13-00517-f004:**
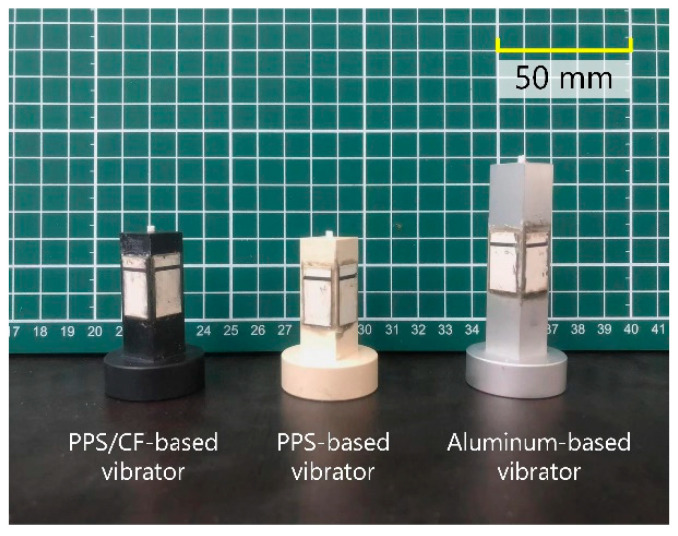
Photograph of the PPS/CF-, PPS-, and aluminum-based vibrators.

**Figure 5 micromachines-13-00517-f005:**
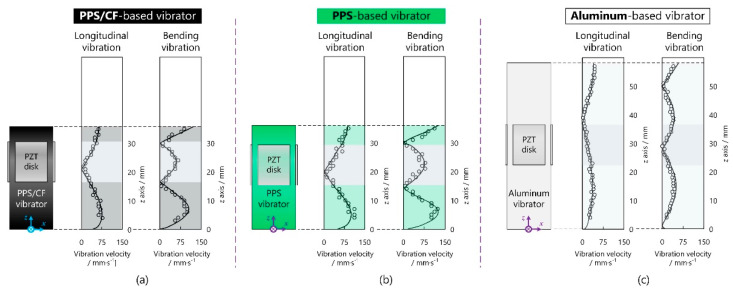
Vibration velocity distributions of PPS/CF-, PPS-, and aluminum-based vibrators.

**Figure 6 micromachines-13-00517-f006:**
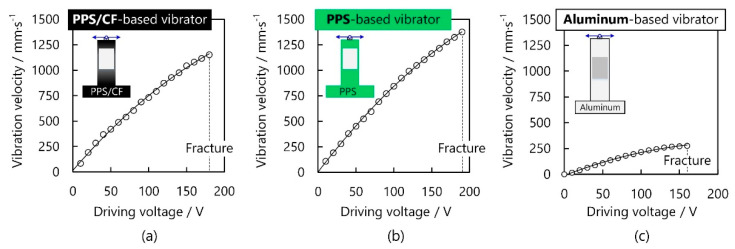
Vibration velocities at the free ends of PPS/CF-, PPS-, and aluminum-based vibrators.

**Figure 7 micromachines-13-00517-f007:**
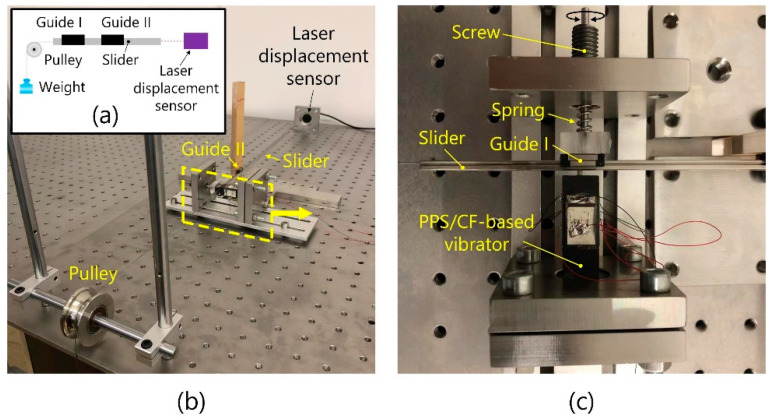
Testbed for measuring the load characteristics of motors. (**a**) Schematic, (**b**) overall figure, and (**c**) zoomed-in image.

**Figure 8 micromachines-13-00517-f008:**
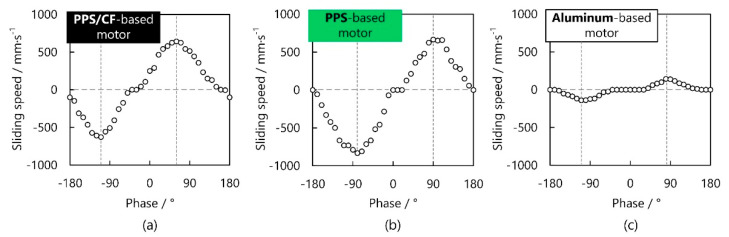
Variation of the sliding speed against the phase applied to the PZT plates exciting longitudinal and bending vibrations for PPS/CF-, PPS-, and aluminum-based motors.

**Figure 9 micromachines-13-00517-f009:**
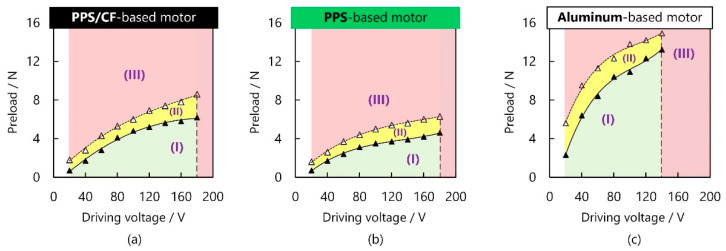
Ranges of the preloads applicable to different voltages for PPS/CF-, PPS-, and aluminum-based motors. The area in green (I) means that the motor worked continuously, the area in yellow (II) means that continuous operation could be obtained but there existed intermittent sticks, and the area in red (III) means the motor could not continuously work.

**Figure 10 micromachines-13-00517-f010:**
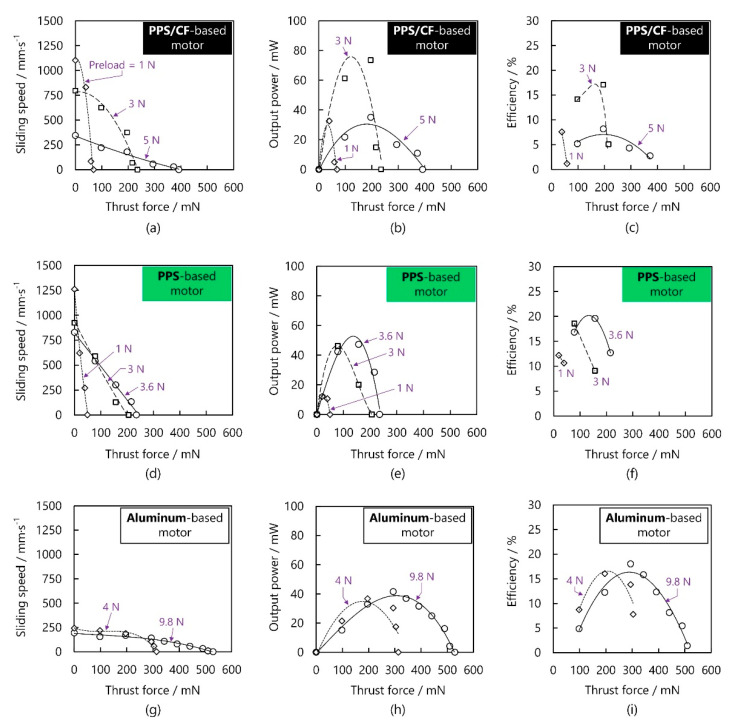
Load characteristics of the PPS/CF- and PPS-based motors at 160 V and those of the aluminum-based motor at 140 V. (**a**) Sliding speed, (**b**) output power, and (**c**) efficiency against the thrust force of the PPS/CF-based motor; (**d**) sliding speed, (**e**) output power, and (**f**) efficiency against the thrust force of the PPS-based motor; (**g**) sliding speed, (**h**) output power, and (**i**) efficiency against the aluminum-based motor.

**Figure 11 micromachines-13-00517-f011:**
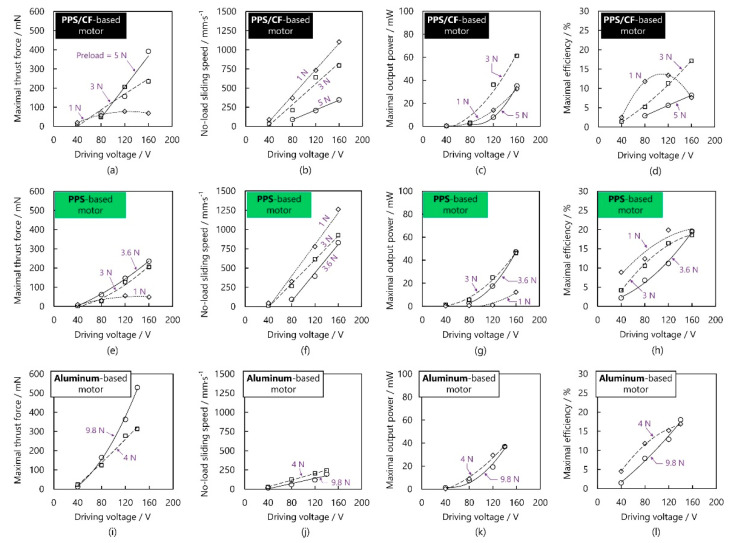
Load characteristics of the PPS/CF-, PPS-, and aluminum-based motors at different voltages and preloads. (**a**) Maximal thrust force, (**b**) no-load sliding speed, (**c**) maximal output power, and (**d**) maximal efficiency against the driving voltage for the PPS/CF-based motor; (**e**) maximal thrust force, (**f**) no-load sliding speed, (**g**) maximal output power, and (**h**) maximal efficiency against the driving voltage for the PPS-based motor; (**i**) maximal thrust force, (**j**) no-load sliding speed, (**k**) maximal output power, and (**l**) maximal efficiency against the driving voltage for the aluminum-based motor.

**Figure 12 micromachines-13-00517-f012:**
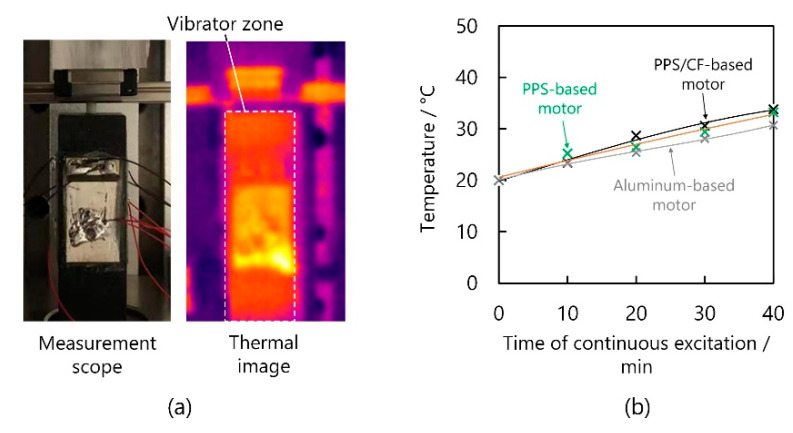
Temperature rise of the motors. (**a**) Measurement scope and thermal image of the PPS/CF-based motor. (**b**) Temperature of the vibrators as functions of time of continous excitation.

**Table 1 micromachines-13-00517-t001:** Material constants of PPS/CF, PPS, and aluminum.

Type	Material	Young’s Modulus/GPa	Density/ × 10^3^ kg/m^3^	Poisson’s Ratio
Anisotropic	PPS/CF	$\left[\begin{array}{cccccc} 11.78 & 2.06 & 9.23 & 0 & 0 & 0\\ & 11.78 & 9.23 & 0 & 0 & 0\\ & & 16.14 & 0 & 0 & 0\\ & & & 3.45 & 0 & 0\\ & sym. & & & 3.45 & 0\\ & & & & & 4.86 \end{array}\right]\;\;$	1.69	-
Isotropic	PPS	3.45	1.36	0.36
	Aluminum	70.3	2.70	0.31

**Table 2 micromachines-13-00517-t002:** Admittance characteristics of PPS/CF-, PPS-, and aluminum-based vibrators.

Indicators (Unit)	PPS/CF-Based Vibrator		PPS-Based Vibrator		Aluminum-Based Vibrator	
	Bending Mode	Longitudinal Mode	Bending Mode	Longitudinal Mode	Bending Mode	Longitudinal Mode
Resonance frequency, *f_r_*/kHz	30.293	30.309	23.090	23.154	54.477	59.518
Anti-resonance frequency, *f_a_*/kHz	30.309	30.334	23.097	23.163	59.663	59.748
Electromechanical coupling factor, *k*	3.24%	4.05%	2.46%	2.78%	7.89%	8.77%
Mechanical quality factor, *Q*	121	146	129	158	387	402
Motional admittance, *A_m_*_0_/mS	0.244	0.440	0.120	0.175	1.109	1.478
Motional resistance, *R_m_*/kΩ	4.098	2.273	8.333	5.714	0.902	0.677
Motional inductance, *L_m_*/mH	2.605	1.742	7.409	6.206	0.934	0.727
Motional capacitance, *C_m_*/nF	0.011	0.016	0.006	0.008	0.077	0.098
Clamped capacitance, *C_d_*/nF	10.3	9.80	10.0	9.70	12.3	11.1

**Table 3 micromachines-13-00517-t003:** Performance comparison between the PPS/CF-based motor with other typical ones.

Performance (Unit)	This Study			[29]
	PPS/CF-Based Motor	PPS-Based Motor	Aluminum-Based Motor	
Material of vibrating body	PPS/CF	PPS	Aluminum	PPS/CF
Vibration modes	B2/L2	B2/L2	B2/L2	B2/B2
Vibrator’s dimension (mm)	45.7 × Φ30	46 × Φ30	68 × Φ30	44 × Φ 8
Vibrator’s weight (g)	19.3	17.4	49.2	2.84
Working frequency (kHz)	30.2	23.1	59.2	4.5
Maximal thrust force (mN)	392	235	529	49.6 (0.248 mNm)
No-load sliding speed (mm/s)	1103	1261	243	150 (30 rad/s)
Maximal output power (mW)	62	47	37	4
Thrust force density (N/kg)	20.3	13.5	10.8	17.6
Power density (W/kg)	3.2	2.7	0.8	1.4

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
