# Peer review of "An Ultrasonic Motor Using a Carbon-Fiber-Reinforced/Poly-Phenylene-Sulfide-Based Vibrator with Bending/Longitudinal Modes"

_micromachines, 2022, doi:10.3390/mi13040517_

Round 1

Reviewer 1 Report

Dear auhtors,

thank you very much and congratulations on the very well written manuscript! I have a few comments only, please find them in the attached pdf.

My main objection is that the characteristics of your motor do not reach those of more than 25 year old state-of-the-art motors that are available commercially. Please find an argument to wipe out this doubt!

Author Response

Response:

Thank you very much for your valuable comments. We have carefully revised this manuscript and hope that this revision can improve the quality of this paper.

  1. Fig. 5, about the position of the PZT plates.

Response:

Thank you very much for your valuable comments.

In Fig. 5(a) and (b), the PZT plates are bonded onto the nodes of the longitudinal modes but they are close to the nodes of the bending modes. In the case of the aluminum-based vibrator, we bonded the PZT plates near the nodes of both the longitudinal and bending modes to efficiently excite these vibrations.

  It has been added in line 244-246. 

  1. Fig. 8, about the deviation of phases

Response:

Thank you very much for your valuable comments.

It is generally found in USMs that the phases corresponding to the maximal speed deviate from the theoretical values for some reasons, such as unperfect fabrication and/or assembly [3].

It has been added in line 294-295.

  1. My main objection is that the characteristics of your motor do not reach those of more than 25 year old state-of-the-art motors that are available commercially. Please find an argument to wipe out this doubt!

Response:

Thank you very much for your valuable comments.

As you mentioned, the PPS/CF-based motor developed in this study does not provide the performance comparable to the commercially-available ultrasonic motors, e.g., Shinsei’s product [38], as some factors, such as the bonding materials, the structure, or the friction materials, are not fully considered, but we assume that this problem does not hamper this work’s contribution of applying the polymer to USMs and showing basic designing procedures. We aim to provide more choice for the vibrating bodies of USMs to broaden the application fields.

It has been added in line 427-433.

Reviewer 2 Report

Linear ultrasonic motor with carbon-fiber-reinforced/poly-phenylene-sulfide (PPS/CF) is developed and analyzed. Measurements of the stiffness characteristics of the PPS/CF sample were made. A numerical and experimental study of the ultrasonic motors was performed and results were compared with the motors made from PPS and aluminium. The main novelty of this investigation is the application of PPS/CF for the stator of the motor however authors do not focus on the microstructure of the PPS/CF and it is not clear what are CF dimensions and pattern.

Remarks and questions

  1. How orientation and density of the CF were controlled during the manufacturing of the stator?
  2. It is not clear what kind of machinery was used to make the stator (3D printing?).
  3. The photo of the PPS/FC must be made using an electronic microscope to show the orientation of the CF and explanation how and why such microstructure allows achieving similar results as the metal stator.
  4. What was the measurement error of stiffness parameters (Section 2). The error distribution must be shown in Fig. 2b.
  5. It is not clear why rectangular shaped PZT plates (Fig.3) are named PZT discs?
  6. How the location of PZT discs on the stator was defined (Fig.3)?
  7. How is the working principle of the motor changing when both electric signals have the same phase (Fig.3). What is the trajectory of the driving foot in this case?
  8. There are ultrasonic piezoelectric motors developed up till now that are driven using a single electric signal? Why the measured speed of the slider is equal to 0 mm/s when the phase was 0 degrees (Fig.8)?
  9. The authors claim that l = 35.7 mm was chosen to reduce weight. But in this case, the frequency L2 and B2 do not coincide so the motor operates in non-resonant mode, and the efficiency of the motor is decreased.
  10. How the length of the PPS and aluminium stator was calculated?
  11. Why speed, power, and efficiency of the motors made from PPS/CF and PPS were not measured under preload 9.8 N?
  12. Why efficiency of the motor made from PPS/FC is similar to the motor made from aluminum, while the quality factor Q of PPS/FC is 3 times lower compared to aluminium.
  13. Comparison of the speed, thrust force and efficiency of the motors mande from PPS/FC, PPS, and aluminium must be made under same preload (for example 4N), and the same voltage of 180-200 V.

Author Response

  1. How orientation and density of the CF were controlled during the manufacturing of the stator?

Response:

Thank you very much for your valuable comments.

The PPS/CF samples are provided by our agent company and they have the properties similar to Toray’s PPS/CF products. We did not make the PPS/CF samples by ourselves. The samples are in cylindrical shapes and the CFs are filled along the axis. So the orientation is controlled by machining the sample along certain directions. The density of CFs is not provided by the company.

Related information has been added in lines 121-123.

  1. It is not clear what kind of machinery was used to make the stator (3D printing?).

Response:

As mentioned above, they are fabricated by conventional machining.

  1. The photo of the PPS/FC must be made using an electronic microscope to show the orientation of the CF and explanation how and why such microstructure allows achieving similar results as the metal stator.

Response:

Thank you very much for your suggestion. We understand that some research regarding the development of new materials need the electronic microscope photos. As the PPS/CF samples are provided, we do not have too much information about the material itself. In the application aspect, please refer to the link of Toray’s product in the manuscript.

In addition, since we are not in the major of material fields, it would be difficult to give a definite answer about the mechanism why PPS and PPS/CF show the acoustic properties different from conventional polymers. However, this property infers the potential to use this material to make USMs, and we aim to confirm this feasibility in this study. Thanks again for your suggestion. We would like to conduct more investigation to clarify the mechanism in further studies. 

  1. What was the measurement error of stiffness parameters (Section 2). The error distribution must be shown in Fig. 2b.

Response:

  In the measurement, coupling materials (liquid) are required between the probe and the sample. Insufficient liquid may lower the magnitude of the received signal; this should be the reason for the measurement error.

  The error bar has been given in Fig. 2(b).

  1. It is not clear why rectangular shaped PZT plates (Fig.3) are named PZT discs?

Response:

  We feel sorry for this typo. It should be ‘PZT plate’ here.

  1. How the location of PZT discs on the stator was defined (Fig.3)?

Response:

  The positions of PZT plates are arranged near the nodes (which means zero vibration velocity) of both the longitudinal and bending modes.

  It has been added in lines 244-246.

  1. How is the working principle of the motor changing when both electric signals have the same phase (Fig.3). What is the trajectory of the driving foot in this case?

Response:

  When the phase becomes zero, the trajectory become a tilting curve instead of an elliptical motion and it cannot achieve the actuation.  

  1. There are ultrasonic piezoelectric motors developed up till now that are driven using a single electric signal? Why the measured speed of the slider is equal to 0 mm/s when the phase was 0 degrees (Fig.8)?

Response:

  When a single signal is applied, it is difficult to control the trajectory. Whether the motor can be driven depends on the motor’s structure. Except the original report of Sashida et al., most ultrasonic motors are driven by two signals. But some piezoelectric motors utilize stick-slip to achieve motion, which is another actuating principle different from that used in this study.

  When the phase is zero, as mentioned above, the trajectory becomes a tilting line, which cannot transfer the vibration into the motion.

  1. The authors claim that l = 35.7 mm was chosen to reduce weight. But in this case, the frequency L2 and B2 do not coincide so the motor operates in non-resonant mode, and the efficiency of the motor is decreased.

Response:

  When l = 35.7 mm, the resonance frequencies are, as shown in Fig. 3(b), both near 30 kHz, so the longitudinal and bending modes are excited on the vibrator. As shown in Fig. 3(e), the longitudinal and bending modes exist on the vibrator alternatively in the time domain. Thus, this motor is a resonant type one, and in this case, the motor is excited efficiently.

  1. How the length of the PPS and aluminum stator was calculated?

Response:

  The PPS and aluminum stator was designed through FEA. The difference is that they have different material constants compared to PPS/CF.

  1. Why speed, power, and efficiency of the motors made from PPS/CF and PPS were not measured under preload 9.8 N?

Response:

  The PPS/CF and PPS-based motor cannot provide continuous motion at 9.8 N because this preload causes the resonance frequencies of two vibration modes to deviate from the original values. Here, we added a figure showing the preload applicable to the motor at different driving voltage.

  Please see Fig. 9 for more information.

  1. Why efficiency of the motor made from PPS/FC is similar to the motor made from aluminum, while the quality factor Q of PPS/FC is 3 times lower compared to aluminum.

Response:

  Thank you for your comment. The power loss of USMs mainly includes the vibration loss and friction loss. The vibration loss is generated when the ultrasound propagates inside the vibrator, and the Q factor can reflect this loss to some extent. While the friction loss exists on the contact surface and it takes the majority of the power loss. Since we adopt the same friction conditions, the friction loss and consequently the efficiency are not greatly different among these motors.

  1. Comparison of the speed, thrust force and efficiency of the motors made from PPS/FC, PPS, and aluminum must be made under same preload (for example 4N), and the same voltage of 180-200 V.

Response:

  Thank you for your suggestion. As shown in Fig. 9, after measuring the voltage and applicable preload, we explored the performance of these motors when they operate continuously and used three levels of preloads to perform the experiments. In addition, the aluminum-based motor cannot work at >140 V because the PZT plate easily fractures here.

  It has been added in lines 301-313.

Round 2

Reviewer 2 Report

The authors made corresponding improvements. The Paper can be accepted for publication.